# Underwhelming Generalization Improvements From Controlling Feature Attribution

**Anonymous**

## Abstract

Overfitting is a common issue in machine learning, which can arise when the model learns to predict class membership using convenient but spuriously-correlated image features instead of the true image features that denote a class. These are typically visualized using saliency maps. In some object classification tasks such as for medical images, one may have some images with masks, indicating a region of interest, i.e., which part of the image contains the most relevant information for the classification. We describe a simple method for taking advantage of such auxiliary labels, by training networks to ignore the distracting features which may be extracted outside of the region of interest, on the training images for which such masks are available. This mask information is only used during training and has an impact on generalization accuracy in a dataset-dependent way. We observe an underwhelming relationship between controlling saliency maps and improving generalization performance.

## 1 Introduction

Overfitting is a common problem in machine learning, particularly when one uses powerful function approximators such as deep neural networks. When training these models with backpropagation, the network will evolve from modelling simple to more complicated functions until it finds salient discriminative features in the data. Once the model has found these, the gradients of the loss do not encourage the model to find other discriminative features in the data, even if they exist (Reed & Marks, 1999). In the classification case, this can be problematic if there exists some distractor feature $x_d$ in the data that is correlated with one of the output classes. This is a common issue in industry data (e.g., medical) where datasets are typically small and there are many confounding variables.

Consider the extreme case in a binary classification problem where in the training distribution there exists a confounding distractor element $x_d$ of the input data such that for $D_{train}$, $p(y = 1|x_d) = 1$, while in the validation distribution $D_{valid}$, $p(y = 0|x_d) = 1$ (Figure 1). In this scenario, predicting using $x_d$ is easier than predicting using the true features that denote class membership and a classifier trained on $D_{train}$ with traditional classification loss would predict the incorrect class with 100% probability on $D_{valid}$. This is a textbook example of overfitting (Goodfellow et al., 2016; Reed & Marks, 1999). The existence of these overfit features is the motivation behind methods seeking to learn domain-invariant representations (Ganin & Lempitsky, 2014; Fernando et al., 2014), and is a common problem with real-world data (Badgeley et al., 2019; Zhao et al., 2019; Young et al., 2019).

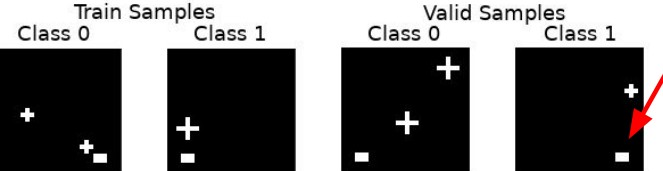

Figure 1: Example images from $D_{train}$ and $D_{valid}$ from both classes. In both distributions, cross size can vary between samples. In $D_{train}$, two crosses (denoting class 0) are always accompanied by a box $x_d$ in the bottom right-hand corner, while a single cross (denoting class 1) is always accompanied by a distractor in the bottom left-hand corner. In $D_{valid}$, the relationship between classes and crosses remains the same, but the logic governing the location of the distractor is reversed. The distractor is indicated with a red arrow.

In this paper, we explore the utility of various methods that allow one to use a mask on the input data to guide the network to avoid predicting from the defined region and penalize the network for attributing a prediction to a distractor. We present a synthetic dataset that encourages all models tested to overfit to an easy to represent distractor instead of a more complicated counting task. We present a novel "activation difference" (*actdiff*) regularizer which mitigates this behaviour directly. We also present a method where we train an autoencoder/UNet to reconstruct a masked version of the input, indirectly controlling feature representations used for classification. We compare these methods with the recently-proposed *gradmask* (Simpson et al., 2019b), and present an expanded analysis of this algorithm's behaviour. All code for this paper, and this dataset, are available here: https://github.com/bigtrellis2222/activmask.

We compare the real-life performance of these methods on open medical datasets with traditional classifiers, and demonstrate the differences in their feature attributions using saliency maps. Finally, we describe a medical dataset curated from two openly-available X-ray databases, and describe how samples can be drawn from each to generate a dataset biased by a site-diagnosis correlation inspired by previous work (Zhao et al., 2019). We demonstrate that, similarly to our synthetic datasets, classifiers are likely to predict using features unrelated to the task, and demonstrate that the proposed methods do mitigate this and often successfully refine the saliency maps to focus on the correct anatomy. However they do not consistently prevent overfitting.

## 2 RELATED WORK

It is a well-documented phenomenon that convolutional neural networks (CNNs), instead of building object-level representations of the input data, tend to find convenient surface-level statistics in the training data that are predictive of class (Jo & Bengio, 2017). Previous work has attempted to reduce the model's proclivity to use distractor features by randomly masking out regions of the input (DeVries & Taylor, 2017). By randomly removing information from the inputs to the network, this method helped the network learn representations that aren't dependent on single feature types in the image. However, this regularization approach gives no control over the kinds of representations learned by the model.

Recently, the Gradmask (Simpson et al., 2019b) and CARE methods (Zhuang et al., 2019) both proposed to control feature representations by penalizing the model for utilizing gradients outside of regions of interest. CARE was additionally designed to deal with class imbalances by increasing the impact of the gradients inside region of interest of the under-represented class.

In contrast to these two methods, we propose a new method which does not work with a saliency map, which can be noisy due to the ReLU activations allowing irrelevant features to pass through the activation function (Kim et al., 2019). Instead, this method operates directly on the activations themselves, encouraging the model to produce similar activation patterns in the presence of, and absence of, information outside of the region of interest.

## 3 METHODS

**Actdiff Loss:** To mitigate the effect of $x_d$, we propose to explicitly regularize the network to ignore $x_d$ at test time by minimizing the distance between the network activations of the model when presented with a full input image and one where the information outside of some mask on the image has been corrupted. The model is only directly trained on unmasked examples. This encourages the network to build features which appear inside the masked regions even though it always sees the full image during training. The method requires having access to masks drawn by an expert who can distinguish between interesting and non-interesting discriminatve features, as is often the case in medical imaging. The *actdiff* regularization term is

$$\boldsymbol{\mathcal{L}_{act}} = \frac{1}{n} \sum_{l=1}^{n} ||o_l(\mathbf{x}_{masked}) - o_l(\mathbf{x})||_2, \tag{1}$$

where $o_l(\mathbf{x})$ are the pre-activation outputs for layer $l$ of the $n$-layer encoder $f(x)$ when the network is presented with the original data $\mathbf{x}$, and $o_l(\mathbf{x}_{masked})$ are the pre-activations outputs for layer $l$ when presented with masked data $\mathbf{x}_{masked}$. We call this the *actdiff* penalty. $\mathbf{x}_{masked}$ should be constructed by randomizing the indices of all pixels that fall outside of the mask, destroying any spatial information available in those regions of the image, but retaining the distribution of intensities

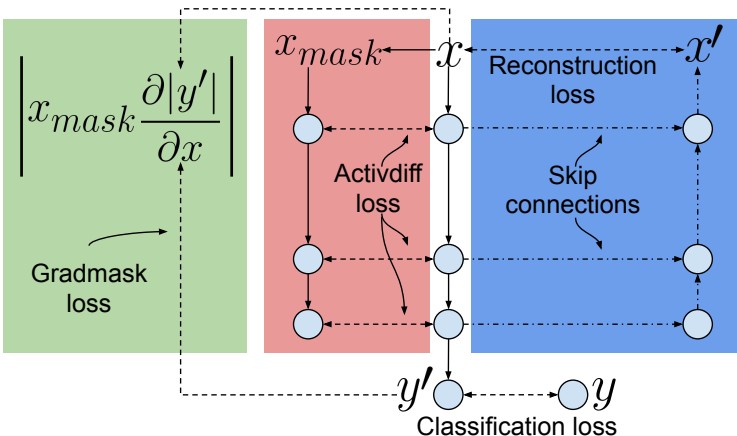

Figure 2: Schematic of the model used in all experiments (alongside an 18-layer ResNet). The *actdiff* penalty was only applied to the encoder path of the model. The reconstruction path (post classification) was optionally used when a reconstruction was requested of the model. Skip connections were optionally employed in the style of UNet. Both of these optional paths are denoted using alternating dashed lines. All losses are denoted using standard dashed lines.

found in the data. Furthermore to retain important context around a masked region, we always dilate the mask by a set number of pixels. This loss has previously been used in style transfer to transform a random noise image into one containing the same layer-wise activations as some target image Gatys et al. (2015).

**Reconstruction Loss:** The pressure from $\mathcal{L}_{act}$ to not represent aspects of the input data $x$ can be balanced with a reconstruction term that encourages the model to represent the data. Concretely, we can employ an auto-encoder architecture $\hat{\mathbf{x}} = g(f(\mathbf{x}))$, where $f$ is a convolutional encoder and $g$ is a deconvolutional decoder, and require the model to minimize the reconstruction loss $\mathcal{L}_{recon} = \frac{1}{m} \sum_{j=1}^{m} ||\mathbf{x}^{(j)} - \hat{\mathbf{x}}^{(j)}||_2$, where $j$ indexes the pixel-wise difference between the input $x$ and reconstruction $\hat{x}$. In our experiments a reconstruction term is helpful for guiding the network toward building useful representations of the data while employing $\mathcal{L}_{act}$. All auto-encoder models presented in this paper employed this loss term. Furthermore, the reconstruction task can be used to indirectly control feature representation learning by asking the model to reconstruct a masked version of the input given the full input $\hat{\mathbf{x}}_{masked} = g(f(\mathbf{x}))$ and minimizing $\mathcal{L}_{recon\_masked} = \frac{1}{m} \sum_{j=1}^{m} ||\mathbf{x}_{masked}^{(j)} - \hat{\mathbf{x}}_{masked}^{(j)}||_2$, which we used as a replacement for $\mathcal{L}_{recon}$.

**Gradmask Loss:** Gradmask is a recently proposed Simpson et al. (2019b) method for controlling which regions of the input are desirable for determining the class label using saliency maps. Saliency maps, or "input feature attribution", can be calculated as $\frac{\partial |\hat{y}_i|}{\partial \mathbf{x}}$ for each input $x$ (Zeiler & Fergus, 2013; Simonyan et al., 2014; Lo et al., 2015). In these experiments we minimize the 'contrast' saliency between healthy and non-healthy classes (labels $y_0$ and $y_1$ respectively), as we expect that input variance which increases the distinction between the two classes leads to overfitting and is what we want to regularize. Therefore, we minimize

$$\mathcal{L}_{grad} = \sum_{\mathbf{x} \in D} \mathcal{L}_{clf} + \left| \frac{\partial |\hat{y}_1 - \hat{y}_0|}{\partial \mathbf{x}} \cdot (1 - \mathbf{x}_{seg}) \right|_2, \qquad (2)$$

where $\hat{y}_0$ and $\hat{y}_1$ are the predicted outputs for our two classes and $(1 - \mathbf{x}_{seg})$ is a binary mask that covers everything outside the defined regions of interest.

**Control Experiments:** For many experiments, we evaluated the effect of simply training a model using masked data (and evaluating it using unmasked data). We call these experiments *Classify Masked*. We present this as a naive alternative to *actdiff*. Additionally, we evaluated what would happen if our auto encoder models were asked to reconstruct a masked version of the input ($\mathcal{L}_{recon\_masked}$ above). These experiments are reported as *Reconstruct Masked*.

## 4 SYNTHETIC DATASET

**Method:** To evaluate the proposed methods for combating overfitting in the presence of a distractor variable, we generated a dataset following the description provided earlier (Figure 1) with 500 training, 128 validation, and 128 test examples respectively. The position of the distractors was perfectly correlated with class label in the training set and the logic governing this relationship was inverted for the validation and test sets. In cases where the model relies on the distractor to make a class prediction, we expected 0.0 AUC for the validation and test sets.

To evaluate the effect of the *actdiff* loss, *gradmask* loss, and the reconstruction penalty during training, we constructed a simple CNN architecture that optionally deconvolved the final layer to generate a reconstruction, or optionally did so in the style of a UNet (Ronneberger et al., 2015), see Figure 2. We additionally tested all non-reconstructing approaches using a simple 18-layer ResNet model (He et al., 2016). As control experiments, we also evaluated classifier performance when simply trained using masked versions of $x$, but evaluated on unmasked examples. All models were trained using Adam for 500 epochs with a learning rate of $10e^{-4}$, batch size of 32, with the weights of our regularizers in the loss term (denoted as $\lambda$) being $\lambda_{act} = 10$, $\lambda_{grad} = 10$, and $\lambda_{recon} = 10$ when applicable. The classification weight was set to 1. Before masking, masks were blurred using a Gaussian filter using a $\sigma = 0.5$, in order for some context to be included around the masked area. For reconstruction, the binary cross entropy loss was used as the input images were binary.

**Results:** The results of all experiments are shown in Table 1, with the architectures that successfully avoid overfitting in bold. All results are the average of 10 random model initilizations and data splits. To determine the effect of *actdiff* on feature representations in the network, we display saliency maps on the validation set during the final epoch of training in Figure 3.

| Experiment Name | Train AUC | Valid AUC | Best Epoch (/500) |
|---|---|---|---|
| Conv AE Classify | $1.00 \pm 0.00$ | $0.06 \pm 0.03$ | $0.00 \pm 0.00$ |
| Conv AE Actdiff | $\mathbf{0.80 \pm 0.26}$ | $\mathbf{0.80 \pm 0.26}$ | $\mathbf{140.10 \pm 156.84}$ |
| Conv AE Gradmask | $0.50 \pm 0.00$ | $0.50 \pm 0.00$ | $44.70 \pm 102.68$ |
| Conv AE Actdiff & Gradmask | $0.50 \pm 0.00$ | $0.50 \pm 0.00$ | $0.00 \pm 0.00$ |
| Conv AE Classify Masked | $1.00 \pm 0.00$ | $0.53 \pm 0.01$ | $127.00 \pm 132.24$ |
| Conv AE Reconstruct Masked | $1.00 \pm 0.00$ | $0.06 \pm 0.05$ | $0.00 \pm 0.00$ |
| CNN Classify | $1.00 \pm 0.00$ | $0.00 \pm 0.01$ | $38.10 \pm 120.13$ |
| CNN Actdiff | $0.50 \pm 0.00$ | $0.50 \pm 0.00$ | $7.30 \pm 23.08$ |
| CNN Gradmask | $0.50 \pm 0.00$ | $0.50 \pm 0.00$ | $0.00 \pm 0.00$ |
| CNN Actdiff & Gradmask | $0.50 \pm 0.00$ | $0.50 \pm 0.00$ | $0.00 \pm 0.00$ |
| CNN Classify Masked | $1.00 \pm 0.00$ | $0.55 \pm 0.02$ | $14.10 \pm 7.36$ |
| ResNet Classify | $1.00 \pm 0.00$ | $0.00 \pm 0.000$ | $0.000 \pm 0.00$ |
| ResNet Actdiff | $\mathbf{1.00 \pm 0.00}$ | $\mathbf{1.00 \pm 0.00}$ | $\mathbf{147.60 \pm 148.59}$ |
| ResNet Gradmask | $0.55 \pm 0.06$ | $0.54 \pm 0.073$ | $243.10 \pm 218.66$ |
| ResNet Actdiff & Gradmask | $0.53 \pm 0.04$ | $0.52 \pm 0.03$ | $143.60 \pm 190.57$ |
| ResNet Classify Masked | $1.00 \pm 0.00$ | $0.73 \pm 0.21$ | $169.80 \pm 178.95$ |
| UNet Classify | $1.00 \pm 0.00$ | $0.05 \pm 0.02$ | $0.00 \pm 0.00$ |
| UNet Actdiff | $\mathbf{1.00 \pm 0.00}$ | $\mathbf{0.90 \pm 0.31}$ | $\mathbf{43.50 \pm 30.06}$ |
| UNet Gradmask | $0.50 \pm 0.00$ | $0.50 \pm 0.00$ | $107.10 \pm 142.43$ |
| UNet Actdiff & Gradmask | $0.63 \pm 0.14$ | $0.60 \pm 0.09$ | $348.00 \pm 160.69$ |
| UNet Classify Masked | $1.00 \pm 0.00$ | $0.54 \pm 0.04$ | $51.40 \pm 96.29$ |
| UNet Reconstruct Masked | $1.00 \pm 0.00$ | $0.01 \pm 0.01$ | $0.00 \pm 0.00$ |

Table 1: Synthetic Dataset Test AUC after 500 epochs, averaged over 10 seeds. Mean and standard deviation presented.

First we demonstrate a CNN overfitting on $D_{train}$, using a simple CNN architecture trained using only the cross entropy loss $\mathcal{L}_{clf} = -\sum_{i=1}^{N} y \log \hat{y}_i$. This model achieves 1.0 AUC on $D_{train}$ and 0.0 AUC on $D_{valid}$. Note that the model is attributing all saliency to the distractor. When the model is trained using masked inputs from $D_{train}$ and trained using $\mathcal{L}_{clf}$, the model performs well on $D_{train}$ but is unsure how to handle the distractor in $D_{valid}$, leading to performance similar to chance.

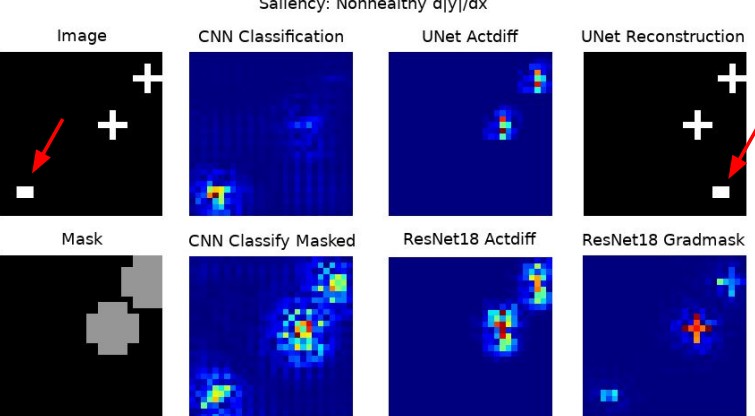

Figure 3: Saliency maps showing the different major behaviours on $D_{valid}$ observed across the models tested on the input image (top left; distractor indicated with a red arrow) with the dialated mask (bottom left). These images correspond to the rows of Table 1. CNN classification demonstrates overfitting, where the saliency is concentrated on the distractor (center top left). CNN Classify Masked demonstrates that the model has not learned to ignore the distractor because it never saw one during training (center bottom left). UNet *actdiff* (center top right) and ResNet18 *actdiff* (center bottom right) demonstrate that the model has successfully learned to ignore the distractor. Note that the network learned to reconstruct the distractor in the location observed during $D_{train}$ (top right, red arrow). The gradmask model fails to ignore the distractor and does not pay equal attention to the two features of interest (bottom right).

A CNN trained using both the classification and *actdiff* loss fails to learn any useful representations, scoring an AUC of 0.5 during train and validation (see "CNN Actdiff" from Table 1). If this model instead a Convolutional Autoencoder (Conv AE) or UNet, and is additionally trained using a reconstruction loss, it can successfully learns to ignore the distractor when classifying the image $x$ ("Conv AE Actdiff" and "UNet Actdiff"). Note in Figure 3 that the UNet reconstruction, taken from the validation set, places the distractor in the location seen in the training set. This implies that the *actdiff* loss guided the model to learn to ignore the distractor before the classification layer, but retained some representation of it in an early layer of the network. Training using reconstruction without Actdiff does not prevent overfitting ("Conv AE Classify" and "UNet Classify"). Experiments with an 18-layer ResNet demonstrate good performance using classification and *actdiff* alone ("ResNet Actdiff").

*Gradmask* never produces a model with good generalizaion performance, with generally unstable training (Figure 7). We suspect this is because the saliency map is always non-zero everywhere on the input, leading to a constant source of noise in the loss function.

The best performing models took many more epochs to reach the optimal solution than would be expected for such a simple dataset. Note the "Best Epoch" for overfit models is misleading as the best performance is chance. We found that models using a reconstruction loss more slowly approach the vicinity of their optimum than a ResNet (Figure 7), and the ResNet model takes longer to each its best epoch (see Table 1).

## 5 SINGLE-SITE MEDICAL DATASET WITH SEGMENTATIONS

**Method:** We applied all previous methods to three medical imaging datasets from the Medical Segmentation Decathlon (MSD) where one task is to detect the presence of an organ in the image, and for each organ we have a segmentation. Classifiers were previously shown to use the incorrect organ when performing this task (Simpson et al., 2019a). We tested our approaches on the liver detection task in CT, cardiac left atrium detection task in MRI, and pancreas detection task in CT. All results are the average of 10 independent seeds, which also lead to independent splits of the data. For each seed 128 training samples, 256 valid samples, and 256 test samples were randomly selected. A sample was a 2D image containing the segmentation. The mask blur factor was $\sigma = 16$

| Experiment Name | Synthetic Test AUC | Liver Test AUC | Cardiac Test AUC | Pancreas Test AUC |
|---|---|---|---|---|
| CNN Classify | $0.35 \pm 0.20$ | $\mathbf{0.87 \pm 0.02}$ | $\mathbf{0.86 \pm 0.13}$ | $\mathbf{0.82 \pm 0.02}$ |
| CNN Actdiff | $0.50 \pm 0.00$ | $0.77 \pm 0.12$ | $0.60 \pm 0.11$ | $0.66 \pm 0.12$ |
| CNN Gradmask | $0.50 \pm 0.00$ | $0.85 \pm 0.01$ | $0.75 \pm 0.20$ | $0.82 \pm 0.02$ |
| CNN Actdiff & Gradmask | $0.50 \pm 0.00$ | $0.79 \pm 0.10$ | $0.50 \pm 0.00$ | $0.66 \pm 0.05$ |
| CNN Classify Masked | $\mathbf{0.74 \pm 0.11}$ | $0.57 \pm 0.05$ | $0.75 \pm 0.07$ | $0.50 \pm 0.01$ |
| ResNet Classify | $0.50 \pm 0.00$ | $0.86 \pm 0.02$ | $0.90 \pm 0.03$ | $0.81 \pm 0.02$ |
| ResNet Actdiff | $\mathbf{0.98 \pm 0.01}$ | $0.86 \pm 0.02$ | $0.88 \pm 0.03$ | $0.81 \pm 0.02$ |
| ResNet Gradmask | $0.71 \pm 0.05$ | $\mathbf{0.89 \pm 0.02}$ | $\mathbf{0.93 \pm 0.01}$ | $0.82 \pm 0.03$ |
| ResNet Actdiff & Gradmask | $0.67 \pm 0.04$ | $0.87 \pm 0.02$ | $0.92 \pm 0.03$ | $\mathbf{0.84 \pm 0.01}$ |
| ResNet Classify Masked | $0.96 \pm 0.03$ | $0.61 \pm 0.03$ | $0.77 \pm 0.08$ | $0.57 \pm 0.05$ |
| Conv AE Classify | $0.50 \pm 0.00$ | $0.78 \pm 0.10$ | $0.76 \pm 0.17$ | $0.77 \pm 0.01$ |
| Conv AE Actdiff | $0.78 \pm 0.24$ | $0.80 \pm 0.02$ | $0.84 \pm 0.08$ | $0.76 \pm 0.02$ |
| Conv AE Gradmask | $0.52 \pm 0.05$ | $0.75 \pm 0.09$ | $\mathbf{0.89 \pm 0.10}$ | $0.77 \pm 0.02$ |
| Conv AE Actdiff & Gradmask | $0.50 \pm 0.00$ | $\mathbf{0.82 \pm 0.01}$ | $0.69 \pm 0.17$ | $\mathbf{0.78 \pm 0.02}$ |
| Conv AE Classify Masked | $\mathbf{0.79 \pm 0.11}$ | $0.59 \pm 0.05$ | $0.69 \pm 0.10$ | $0.51 \pm 0.03$ |
| Conv AE Reconstruct Masked | $0.50 \pm 0.00$ | $\mathbf{0.84 \pm 0.02}$ | $0.82 \pm 0.17$ | $\mathbf{0.81 \pm 0.02}$ |
| UNet Classify | $0.50 \pm 0.00$ | $0.81 \pm 0.07$ | $0.83 \pm 0.17$ | $0.77 \pm 0.09$ |
| UNet Actdiff | $\mathbf{0.95 \pm 0.12}$ | $0.82 \pm 0.02$ | $\mathbf{0.88 \pm 0.03}$ | $0.78 \pm 0.03$ |
| UNet Gradmask | $0.55 \pm 0.07$ | $0.81 \pm 0.02$ | $0.82 \pm 0.13$ | $0.79 \pm 0.02$ |
| UNet Actdiff & Gradmask | $0.63 \pm 0.10$ | $0.78 \pm 0.04$ | $0.78 \pm 0.09$ | $0.76 \pm 0.02$ |
| UNet Classify Masked | $0.74 \pm 0.10$ | $0.58 \pm 0.09$ | $0.56 \pm 0.10$ | $0.53 \pm 0.05$ |
| UNet Reconstruct Masked | $0.50 \pm 0.00$ | $\mathbf{0.87 \pm 0.01}$ | $0.75 \pm 0.20$ | $\mathbf{0.81 \pm 0.03}$ |

Table 2: Test Results (Best Valid Epoch over 500 epochs) on all 3 MSD Datasets. Results are averaged over 10 seeds, and we present the standard deviation.

for the Liver and Pancreas datasets, and $\sigma = 8$ for the cardiac dataset. All images were resized to $100 \times 100$ pixels. These blur values were selected as they provided the best *gradmask* performance, and are higher for the medical datasets when compared with the synthetic dataset experiments as the images used here are much larger ($512^2$ vs $24^2$). All models were trained with an Adam optimizer with a learning rate of $10e^{-4}$ for the Pancreas and Liver datasets, and $4e^{-3}$ for the Cardiac dataset, which were found to be the optimal learning rates using a hyperparameter search. All models were trained with a batch size of 32 and batch shuffling. The weights of our regularizers in the loss term (denoted as $\lambda$) were set to $\lambda_{act} = 1$, $\lambda_{grad} = 1$, and $\lambda_{recon} = 1$, when applicable. The classification loss weight was always set to 1. The reconstruction loss was the mean squared error.

**Results:** We present in all test AUCs for the best valid AUC over 500 epochs of training in Table 2 alongside the experiments from the previous section. For each model, the best-performing (or otherwise notable) configurations are in bold.

For CNN-based models, classification alone gave best performance. In contrast, the ResNet model generally benefited from the addition of *gradmask*, in contrast to the synthetic dataset results. The one notable exception was for the pancreas dataset, where the combination of classification, *actdiff*, and *gradmask* gave the best performance.

The best-performing model at baseline was the ResNet model, which is unsurprising given its superior expressive power over the simple CNN architecture we used for all other experiments. For the Liver and Cardiac datasets, classification with *gradmask* outperformed the baseline and all other models. In the case of the Pancreas dataset, classification with both *actdiff* and *gradmask* performed best. In all cases, classification with *actdiff* alone performed as well as, or worse than, the baseline.

Surprisingly, the auto-encoding models showed the best performance when trained to reconstruct a masked version of the input for the Liver and Pancreas dataset. For the Cardiac dataset, the best performing method was classification with *gradmask* for the Convolutional AutoEncoder and with *actdiff* for the UNet, and each achieve similar performance. This is likely because *actdiff* is too strong of a regularizer for CNN models, so the skip connections in the UNet allow the model to greatly reduce the actdiff penalty in the deeper layers of the encoder. The variance of model performance across seeds is higher if trained with *gradmask* than *actdiff*. Again, training with *actdiff* and *gradmask* outperforms either approach alone in the Liver and Pancreas datasets.

Individual saliency map examples for the Liver, Cardiac, and Pancreas datasets can be found in Figure 9, and the mean saliency over 100 test-set examples for all datasets can be found in Figure 4. In both the the Liver and Cardiac dataset, *actdiff* encourages the ResNet model to focus on the correct

anatomy, but this does not lead to an increase in test AUC performance over baseline. In contrast, *gradmask* less consistently encourages the model to focus its attention on the specified anatomy, but results in consistent test AUC performance improvements relative to baseline. The combination of the two methods ("Actgrad") also produces improved feature attribution in the absence of improved generalization performance. In the Pancreas dataset, we see both *actdiff* and *gradmask* both focus the saliency maps of the model broadly across the anatomy. The *gradmask* and combination *actdiff* and *gradmask* models improved over baseline, but there is no clear reason why this would be true from the saliency maps. We therefore conclude an inconsistent relationship between improved generalization performance and refined saliency maps, where improved saliency does not nessicarily lead to improved performance.

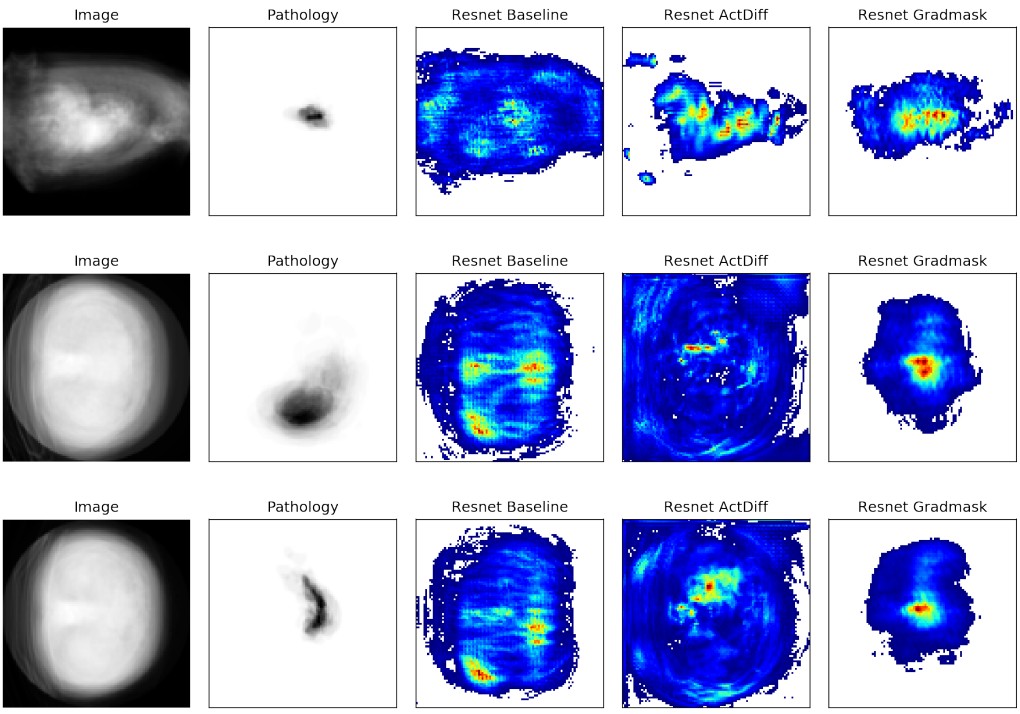

Figure 4: The average saliency map from 100 randomly-selected test-set images in the Cardiac (top), Liver (middle), and Pancreas (bottom) dataset.The pathology column shows the mean non-blurred segmentation for each dataset. All three datasets show clear and consistent changes in the location of the strongest saliencies when using both Actdiff and Gradmask when compared with a normal classifier.

## 6  MULTI-SITE X-RAY DATASET

**Method:** In an attempt to replicate the results of the synthetic dataset in a real world application, we constructed an X-Ray dataset using a combination of the PadChest (Bustos et al., 2019) dataset and the NIH Chestx-Ray8 (Wang et al., 2017) dataset. A site-driven overfitting signal has previously been reported when combining these datasets around the border of the images (Zech et al., 2018). We also observed this in regions far from the lungs (see the mean X-ray from each dataset in Figure 8), and therefore hypothesized we could improve overfitting performance by masking out the edges of the image using a circular mask. We constructed a joint dataset that allowed us to define a site-pathology correlation in the training set, and then produce validation and test set where the reverse relationship is true. In the training set, 90% of the unhealthy patients were drawn from the PadChest dataset and the remaining 10% of the unhealthy patients were drawn from the NIH dataset, and the reverse logic was followed for the validation and test sets. In all splits the classes and site distributions were always balanced, making it tempting for the classifier to use a site-specific feature when predicting the class in the presence of site-pathology correlation. We chose emphysema detection as the detection task, resulting in 998 samples for training, 498 samples for validation and 504 samples for test. All images were resized to $128 \times 128$ pixels. All experiments trained a 18-layer

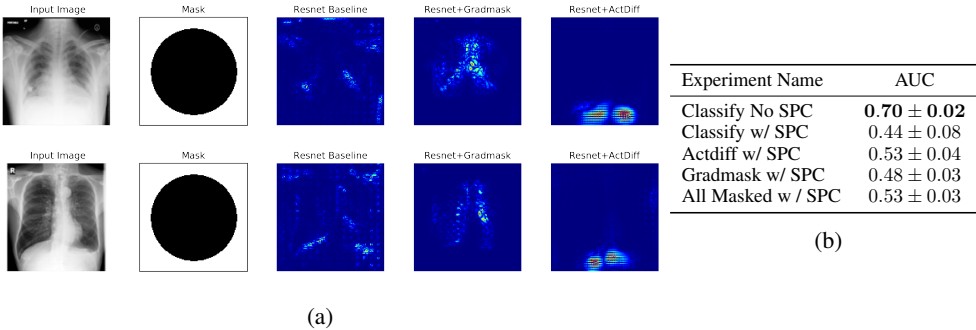

(a)

| Experiment Name | AUC |
|---|---|
| Classify No SPC | $\mathbf{0.70 \pm 0.02}$ |
| Classify w/ SPC | $0.44 \pm 0.08$ |
| Actdiff w/ SPC | $0.53 \pm 0.04$ |
| Gradmask w/ SPC | $0.48 \pm 0.03$ |
| All Masked w / SPC | $0.53 \pm 0.03$ |

(b)

Figure 5: Results on the chest X-ray task. (a) Saliency maps of the different models with different methods to prevent incorrect feature attribution. The task is to predict Emphysema (a lung condition). Two different images from the test set are shown with the masks that were used during training. The top image is a negative example and the bottom positive. (b) Test Results (Best Valid Epoch) using a ResNet on the Chest X-ray task. SPC=site-pathology correlation.

ResNet model using an Adam optimizer with a learning rate of $10e^{-3}$ for 100 epochs. All results were averaged over 10 seeds. We trained a classifier on the same dataset with no site-pathology correlation as a baseline, and compare these results that with the same classifier in the face of a site-pathology correlation of 90%. We train two models using $\mathcal{L}_{act}$ and $\mathcal{L}_{grad}$ with masks for all input images in the training set. We also trained a normal classifier with no $\mathcal{L}_{act}$ and $\mathcal{L}_{grad}$ penalty, but where the masked region was set to zero in the train, valid, and test splits, to obtain an upper bound on the expected performance of our model.

**Results:** See Table 5b. A ResNet trained on the dataset mixed, without a site-pathology bias scores a test AUC of $0.7$, while one trained in the presence of a strong site-pathology bias scores below chance on the test set ($0.44$). Both *actdiff* and *gradmask* improve performance of the model, but only *actdiff* scores above chance, and performs similarly to a model trained with the areas outside of the mask completely removed. However, the saliency maps of the ResNet trained with *actdiff* shows strong feature attribution from outside of the lungs. The model appears to be paying attention to the brightest regions of the image, which might be predictive of the scanner. Since a region of high intensity is available in the center of each X-Ray, *actdiff* when trained with these masks cannot handle this case of overfitting. Models trained with *gradmask* appropriately attribute more saliency to the lungs, but score below-chance level on the test set. Generally, the poor performance of all models in the presence of a site-pathology bias suggests that there is no regional source of the site-bias. This bias likely exists in almost every pixel of the dataset and therefore methods such as *actdiff* or *gradmask* are not well-suited to controlling overfitting in these scenarios.

## 7 CONCLUSION

We hypothesized that poor generalization performance could be partially attributable to classifiers exploiting spatially-distinct distractor features, and proposed the *actdiff* regularizer that prevents this behaviour on a synthetic dataset. We compare the performance of this method against previously-proposed methods operating on saliency maps and demonstrate that the methods influence feature construction and generalization performance in a dataset-dependent manner. We conclude that while our methods successfully control the features constructed from the data, and solve the overfitting problem in a synthetic setting where the distracting feature is spatially distinct from the discriminative features, in real data we found no evidence of a spatially-distinct signal that can be reliably removed to mitigate overfitting. We now doubt the validity of using saliency maps for diagnosing whether a model is overfit because improving them does consistently improve generalization. Improved generalization performance observed in saliency-map based approaches may be more due to the fact that these approaches add useful noise into the updates, similarly to cutout (DeVries & Taylor, 2017). We leave this conjecture to future work.

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

APPENDIX

## A    MASK REQUIREMENT

*Actdiff*'s requirement for hand-drawn masks can be a detriment in practice as they are costly to acquire from human experts. To determine whether *actdiff* has applicability in the setting where only a subset of the training set has masks, we repeated our experiments detailed above, retaining either 20%, 40%, 60%, 80%, or 100% of the masks in the training set. We analyzed the resulting final test AUC (Figure 6a) and the number of epochs (Figure 6b) required to reach this level of performance on the two best-performing *actdiff* models: the UNet and the ResNet18 model, averaging across 5 seeds. In general, the ResNet18 model appears to be more robust to missing masks, although across datasets, there does not seem to be a direct correlation between more masks and better performance. In fact, the addition of more masks can decrease performance, suggesting that the quality of the masks used is more important than the quantity (Figure 6a). There was no consistent effect of the number of masks used during training and the number of epochs required to reach the best epoch. We suspect having a small set of very precise masks is sufficient to guide the model toward developing good representations of the anatomy given that more compute time is available.

## B    ARCHITECTURE OF THE CNN, AUTOENCODER, AND UNET MODEL

The encoder of the AutoEncoder and UNet model was shared with the CNN model and was 4 layers deep, with each layer consisting of a double convolution (kernel size of 3 and a stride of 1). All predictions were made off of the deepest layer of the network. The number of input channels was 16 for the synthetic dataset and 64 for the medical datasets, and doubled for each subsequent layer. All activations for *actdiff* were saved before applying the ReLU activation during the forward pass. During reconstruction, a sigmoid activation was optionally applied to the output to assist in the binary output case (for the synthetic dataset). In the decoder path of the autoencoding models, upsampling was applied using bilinear interpolation before each double convolution.

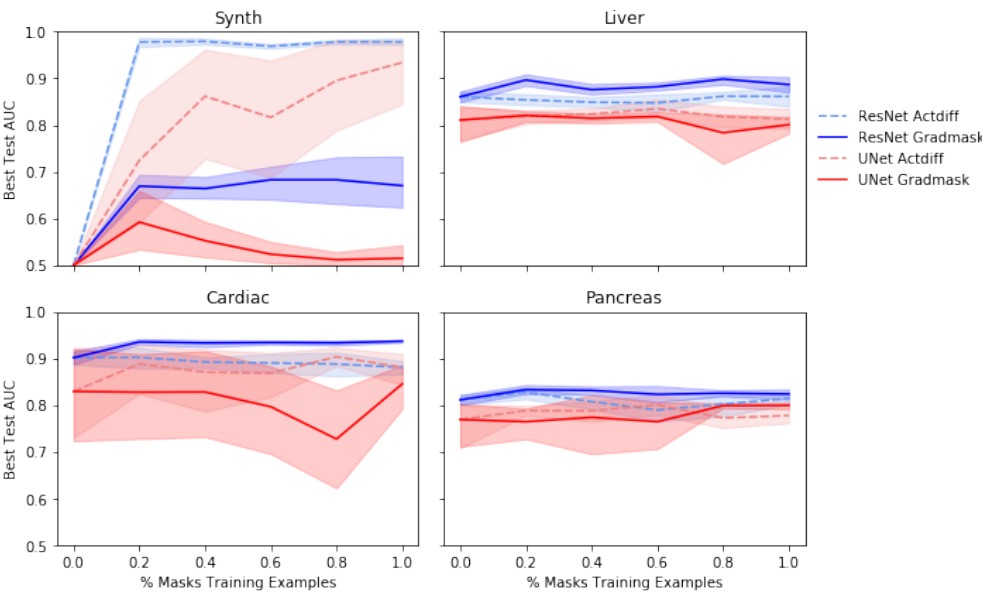

(a) Mean and standard deviation of the test AUC (as selected by the valid AUC) across 10 seeds for all experiments, under the condition that only some percentage of the masks are available. Only shown for the two best-performing Actdiff models (UNet and ResNet18).

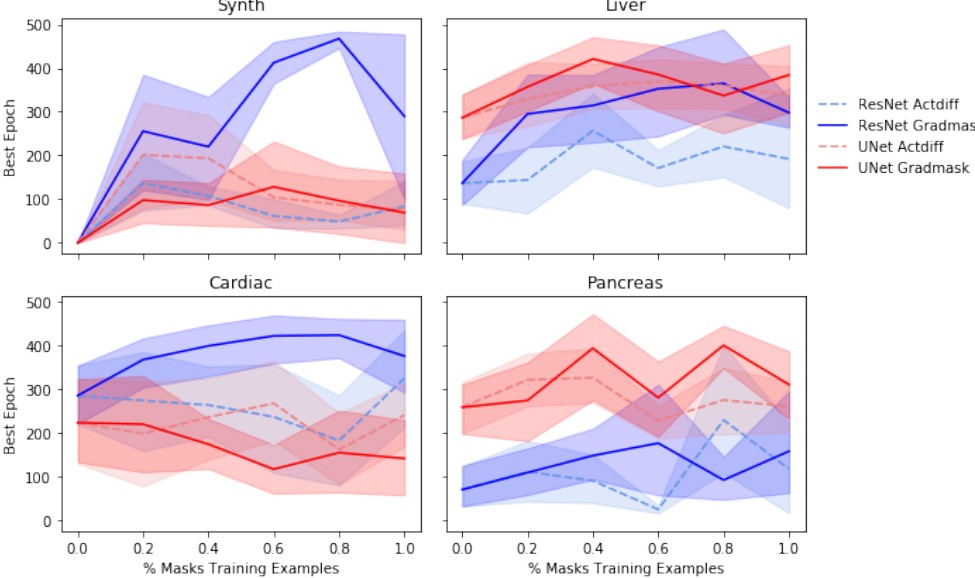

(b) Mean and standard deviation of the best epoch (out of 500) based on the valid AUC across 10 seeds for all experiments, under the condition that only some percentage of the masks are available. Only shown for the two best-performing Actdiff models (UNet and ResNet18).

Figure 6: Results of the maximum masks experiments. (a) Best Test AUC for the best Valid AUC for each of the maximum masks conditions. (b) Best valid Epoch for each of the maximum masks conditions.

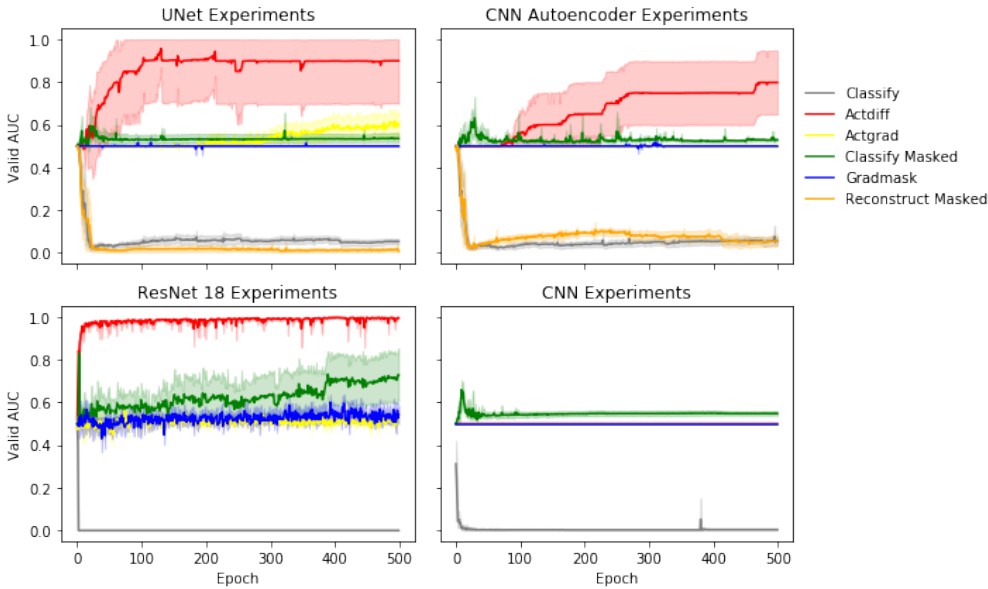

Figure 7: Line plots showing the Valid AUC for each of the 500 epochs during training for all models on the synthetic dataset. We can see that training models with masked data has no substantial benefit on the validation set, models simply trained to classify (or, in addition, reconstruct a masked version of the input) overfit early in training, *gradmask* models fails to train, and *actdiff* surpasses the performance in all cases where it is effective (i.e., not the CNN model).

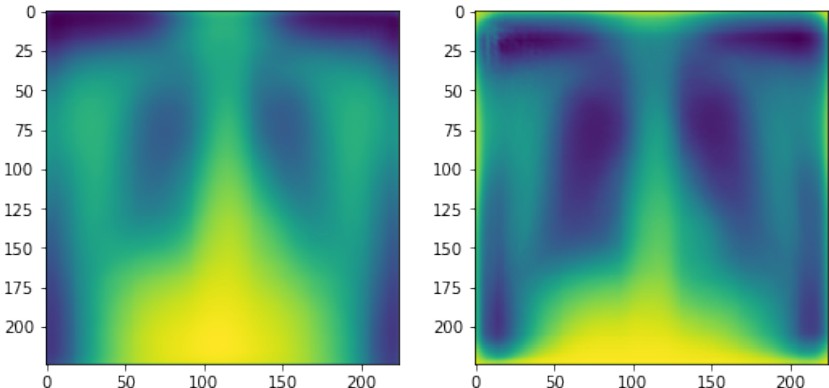

Figure 8: Mean resized X-Ray from the NIH dataset (left) and PadChest dataset (right). There are clear differences in the site distributions that are obvious around the edges of the image.

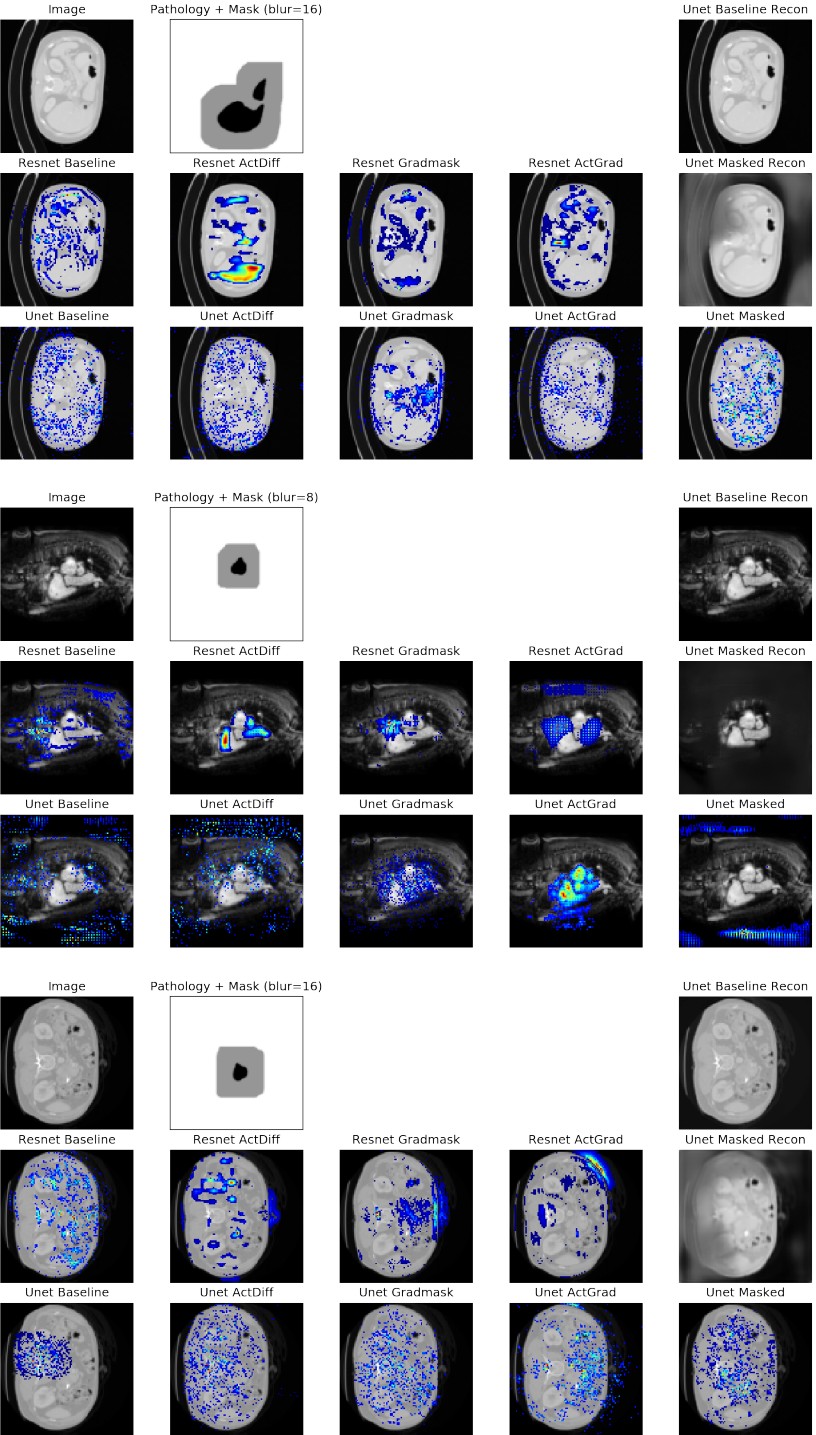

Figure 9: Saliency maps showing where the model attributes areas of the visual input space to the prediction made by the network for the Liver detection (top), Caridac Left Atrium detection (middle), and Pancreas (bottom) datasets. The top 10% of gradients are shown in each image for visualization. The top left image shows the raw input, and to its right is the anatomy segmentation before and after blurring. From left to right along the bottom, the ResNet model outputs are shown in the second row and the UNet results are shown in the third row, for the baseline classification model, ActDiff, Gradmask, and ActDiff & Gradmask ("ActGrad"). The rightmost column shows outputs specific to the UNet reconstructions: the top image shows the standard reconstruction, right middle image shows the output of the Reconstruct Masked task, and bottom image shows the feature attribution of the Reconstruct Masked model.

