# OpenReview forum: "Underwhelming Generalization Improvements From Controlling Feature Attribution"
_ICLR.cc/2020/Conference — Reject_

### Official Review · AnonReviewer2 · 2019-10-21
**Official Blind Review #2**

**Rating:** 3

**Review:**

This paper attempts to tackle the overfitting problem due to the focusing on the distracting information. By utilizing the dataset with masks, the authors propose a simple method to ignore the distracting features. By designing actdiff loss and reconstruction loss, the authors demonstrate that classifiers are likely to predict using features unrelated to the task and their losses can mitigate this problem.

Pros:
This kind of study can make us understand deeply what is going on in deep learning and thus to make it better, which shows this work’s high significance.

Cons:
1.	The presentation of this paper is not clear. For example, in Table 1, the experiment name is so confusing. The authors mention that “Conv AE Actdiff” is the one using both actdiff loss and reconstruction loss. Then what does “Conv AE Reconstruct Masked” mean? Does Conv AE means they implicitly contain reconstruction loss? The authors should elaborate them clearly. The corresponding double quotation marks should be revised. Figure 3 is also confusing. At first, I thought different lines represent different input data (one without mask and one with mask) and different columns represent different methods. However, this is not the case. Furthermore, the fourth column is more confusing. I do not see any red circles in the fourth column. Why the reconstruction of distractor is far away from the original one. Does the right top picture show the advantages of UNet Reconstruction or its drawbacks?
2.	In fact, the training data with masking is not sufficient (may only available in some medical images). It is hard to utilize these data to generalize to various other tasks including large amounts of images without masking.
3.	From the experiments on real tasks, I barely see improvements by the proposed method, which makes the conclusion unconvincing.



**Experience Assessment:**

I have read many papers in this area.

**Review Assessment: Checking Correctness Of Derivations And Theory:**

I carefully checked the derivations and theory.

**Review Assessment: Checking Correctness Of Experiments:**

I carefully checked the experiments.

**Review Assessment: Thoroughness In Paper Reading:**

I read the paper thoroughly.

---

> ### Author Response · Authors · 2019-11-14
> **We're happy to see you're as underwhelmed by the improvements in generalization performance as we were. Please see Figure 4 for evidence that these regularizers are having powerful control over the saliency maps while barely changing generalization performance.**
>
> >  The presentation of this paper is not clear. For example, in Table 1, the experiment name is so confusing. The authors mention that “Conv AE Actdiff” is the one using both actdiff loss and reconstruction loss. Then what does “Conv AE Reconstruct Masked” mean?
>
> Thank you for pointing out this omission! We have clarified these issues in a dedicated section (Page 3; “Control Experiments”).
>
> >  Does Conv AE means they implicitly contain reconstruction loss? The authors should elaborate them clearly. The corresponding double quotation marks should be revised.
>
> Thank you for this suggestion, we have clarified this point on Page 3 in the Reconstruction Loss section: “ In our experiments a reconstruction term is helpful for guiding the network toward building useful representations of the data while employing $\mathcal{L}_{act}$. All auto-encoder models presented in this paper employed this loss term.”.
>
> >  Figure 3 is also confusing. At first, I thought different lines represent different input data (one without mask and one with mask) and different columns represent different methods. However, this is not the case. Furthermore, the fourth column is more confusing. I do not see any red circles in the fourth column. Why the reconstruction of distractor is far away from the original one. Does the right top picture show the advantages of UNet Reconstruction or its drawbacks?
>
> Thank you for pointing out the challenges with Figure 3. The individual images in the figure are meant to be read independently, there is no row or column logic, hence their individual titles that correspond to the results in the tables. We have clarified this in the caption: “These images correspond to the rows of Table \ref{table:synthendresults}”. We have clarified the meaning of the red arrows and fixed the mention of the circle. We also elaborate on the observation about the distractor flipping locations in the reconstruction .
>
> >  In fact, the training data with masking is not sufficient (may only available in some medical images). It is hard to utilize these data to generalize to various other tasks including large amounts of images without masking.
>
> We agree that the requirement of masks limits the application of this method, but even more importantly, our results suggest that controling feature attribution in this method is unlikely to produce improved generalization performance in general, so this becomes a moot point. As an aside, we have some preliminary results suggesting that the method works if only some of the inputs are masked (see the Appendix, Section B).
>
> >  From the experiments on real tasks, I barely see improvements by the proposed method, which makes the conclusion unconvincing.
>
> We are confused by this comment. Our conclusion is that controlling saliency (which our results clearly show) has a negligible impact on improving generalization. In other words, it appears you agree with our assessment of the results.

---

### Official Review · AnonReviewer3 · 2019-10-22
**Official Blind Review #3**

**Rating:** 3

**Review:**

This paper considers how we can train image classification models so that they can ignore task irrelevant features.
For this purpose, the authors considered a situation where task relevant parts of the images are annotated as masks by the human experts.
The authors then proposed using Actdiff loss, Reconstruction loss, and Gradmask loss that are designed to suppress the effect of irrelevant features.

I think the problem considered in this paper is interesting and important.
As the authors pointed out, the medical images taken from different hospitals may contain hospital-specific features which is irrelevant to the targeting task.
Thus, we need a way to train image classifiers that can ignore such irrelevant features.

My major concern on this study is the experiments.
The authors mention that part of their methods sometimes did not work (see below).
This makes the effectiveness of the proposed losses a bit questionable.
If I understad correctly, in practice, the users need to tune the weights of several losses carefully until they can obtain a good model.
If this is the case, I am not very sure if the proposed losses are essential, or tuning a right weight can occasionally provide good models.

* (Sec4) "Gradmask proves to be too powerful a regularizer for this task, and never produces a model with good generalization performance."

* (Sec5) "For the Cardiac dataset, the best performing method was classification with gradmask for the Convolutional AutoEncoder and with actdiff for the UNet, and each achieve similar performance."

* (Sec5) "The gradmask and combination actdiff and gradmask models improved over baseline, but there is no clear reason why this would be true from the saliency maps."

* (Sec6) "Both actdiff and gradmask improve performance of the model, but only actdiff scores above chance, and performs similarly to a model trained with the areas outside of the mask completely removed."

### Updated after author response ###
The authors have tried to address my concern by re-running the experiments, which I greatly appreciate.
However, the effectiveness of the proposed approach is not convincing enough.
I expect the authors to design the more effective experiments in the future version.
It would be great if the authors can clarify and demonstrate which regularization is helpful under which circumstances and why, and when it is not.

**Experience Assessment:**

I do not know much about this area.

**Review Assessment: Checking Correctness Of Derivations And Theory:**

N/A

**Review Assessment: Checking Correctness Of Experiments:**

I assessed the sensibility of the experiments.

**Review Assessment: Thoroughness In Paper Reading:**

N/A

---

> ### Author Response · Authors · 2019-11-14
> **Appreciate your guidance! Please see our new Figure 4 for evidence that our regularizers change saliency, but not generalization performance.**
>
> Thank you for your comments on our methodology. We agree that it would have been better to run an extensive hyperparameter search for all models presented, but we did not have the time or computational resources available to support such a search. As a proxy, we tuned the hyperparameters on baselines and then applied those hyperparameters to our newly presented model. I.e., the learning rate was selected using a normal classification model with no regularization, and then that learning rate was applied to all matching regularized models. We would also refer you to our response to reviewer 1, who covers many of the same concerns.
>
> In the case of the blur, we tuned this using gradmask, since it is the less-novel of the approaches presented here, and we wanted to treat it as a baseline relative to actdiff. We suspect that if we properly tuned the hyperparameters of the models presented, we would get a small boost in performance for our actdiff and gradmask models, but we find it implausible that the performance boost would be so great that it would undermine the conclusion of our paper, namely, that large changes in feature attribution on the inputs (Figure 4, now updated with many more examples from the test set) of a classification network change the resulting performance of a model in underwhelming ways. In particular, we do not believe the final X-ray experiment results would change from “not working” to “working”.
>
> We agree that these results are disappointing, and would love to see future work solve this problem, but we find it unlikely that it will be solved by controlling feature attribution on the network inputs.

---

### Official Review · AnonReviewer1 · 2019-10-25
**Official Blind Review #1**

**Rating:** 3

**Review:**


Summary
---

(motivation)
CNN image classifiers tend to overfit to distractor patterns.
Perhaps these patterns are spatially local, such that in most images signal is at one
location while the noise models tend to overfit to is somewhere else.
If so, then generalization should improve if models are given additional
supervision (i.e., a mask identifying salient regions) that specifies where the signal is and is not.
This paper designs the Actdiff loss to realize this intuition.

(approach)
Actdiff:
1) The Actdiff loss requires a mask that highlights areas of the input
image which have signal and not distractor regions. It extracts features from
the original input image and its masked version then encourages the two features
to be similar at every layer of the CNN using an L2 loss.

Actdiff is compared to 5 other methods including a reconstruction loss and Gradmask (previous work).

(evaluation - synthetic dataset)
A synthetic dataset is constructed for a simple binary classification task based on the presence of simple shapes.
Two patterns can predict the correct class at train time, but one of the patterns is removed at test time so additional information (masks in this case)
is required to specify which pattern the classifier should use.
All the losses except Actdiff achieve at best 50% accuracy on the val set, but Actdiff gets 80% or more accuracy on 3 of 4 tested model variations.
This shows that Actdiff can effectively introduce the relevant masking information.

(evaluation - Medical Segmentation Decathalon)
This dataset provides 3 segmentation tasks (Liver, Cardiac, Pancreas), including ground truth masks for those images.
All 6 methods outperform all the others at least some of the time.
The conclusion is that adding mask information using Actdiff doesn't improve segmentation performance.

(evaluation - Multi-Site dataset)
A final task tries to construct another synthetic dataset out of real X-ray images
collected at two different places.
The train set is largely from one place and the test set is mostly from the other place, and masks are constructed so Actdiff can try to eliminate this bias.
Actdiff causes a negligible increase in performance.

The paper concludes that signals CNNs tend to fit to in this type of data are not very spatially distinct.


Strengths
---

There is some novelty in the approach. The actdiff loss makes sense and masking + activation mapping have not been tried together before to my knowledge.

Experiments follow a logical progression, starting by verifying the idea on a synthetic dataset, then moving to real data, and then evaluating on a half-synthetic dataset designed to debug the approach.

Experiments average over many random initializations.

The paper embraces its negative result.


Weaknesses
---

The approach is not very compelling to me:
* Implicit in this paper is that any information outside the mask is a distractor and any information inside is not a distractor. Why should the particular masks chosen for the experiments have this property? How can an expert know which features a model will find useful?

The paper's novelty is somewhat limited. The idea of regularizing using saliency maps has been explored and even applied to medical data like the MSD used here in Gradmask (one of the strong baselines this paper compares to). Activation matching is also common (e.g. [1]), though it has not been combined with masking before.

While the weights applied to the various losses are provided, it's not clear how they were tuned. In this case there may be lots of competing losses, so it's important to tune the weights somehow to ensure the tradeoff between losses is optimal.

The results (and the conclusions) suggest Actdiff is not very effective at increasing generalization. Table 2 reports test results on all datasets. In that table, each loss outperforms all the other losses in at least two cases (a case is a model-dataset pair).

[1]: Gatys, Leon A. et al. “A Neural Algorithm of Artistic Style.” ArXiv abs/1508.06576 (2015): n. pag.


Missing experiments:

* In the Multi-Site experiment, compare to what happens when the circular mask is applied to all images and not just those from one site. This is a necessary control to be sure that any benefits from masking are due to domain transfer and not other regularization effects. Either conclusion could be useful, but it would be nice to know.

Presentation weaknesses:

* In the synthetic dataset the model cannot tell the difference between correct and incorrect signals at train time. Therefore, I think there's no way for some of the baselines (plain classifier, autoencoder) to generalize correctly. Is that right? If so, it should be clear that comparisons to these baselines are not fair when discussing the synthetic evaluation in section 4.

Missing details / points of clarification:

* What is the Conv AE? I assume it is a CNN based autoencoder of some sort. A detailed description of the non-standard architectures would be useful for reproducibility, though probably only in the appendix.

* What are the lambda hyperparameters? I assume these are weights on the corresponding loss terms, but this is never made explicit.

* How does f(.) relate to the function o_l(.)? Is o_l(.) an intermediate step in f(.)?

* The MSD dataset is not clearly described. Is this a classification dataset where classes are different diseases? What do the ground truth masks capture?

* I think only Conv AE and UNet contain reconstruction losses. This presentation is a bit confusing since reconstruction loss was presented as another loss and it shows up in the tables implicitly based on the architecture being compared.


Suggestions
---

* This paper would be a bit more convincing if it started with a concrete example of the problem illustrated on some dataset (e.g., maybe an example from Gradmask). That may also help drive intuitions later on in the paper.


Preliminary Evaluation
---

Clarity: The paper is fairly clear.
Quality: Quality is mixed. Lots of relevant experiments are reported but they don't support clear conclusions and I'm not sure how well the models were tuned.
Originality: There is some novelty, but it is limited as discussed above.
Significance: I see limited significance.

For me this paper requires special scruitiny because it presents a negative result. Here are some factors that come to mind when thinking about whether to publish a negative result:
* Is the approach compelling? - This approach is not very compelling (e.g., comments about limited novelty and lack of concrete examples to boost intuition).
* Are the experiments thorough? - The experiments could be significantly more thorough (e.g., comments about tuning lambda).
* Will readers learn something useful? - This paper may help researchers trying to leverage similar intuitions, but it won't be very useful outside this audience.
* Does the paper present experiments that promote deeper understanding of why the approach failed? - This paper makes significant reasonable steps in that direction with sections 4 and 6, but I was still a bit dissapointed with the conclusions of these sections.
* Does the paper discuss alternative approaches that were investigated? - Many alterantive approaches were considered and their performance reported.

Overall I think this paper is close but fails to meet the bar because it does a bit worse than expected on most criteria above.


**Experience Assessment:**

I have published one or two papers in this area.

**Review Assessment: Checking Correctness Of Derivations And Theory:**

N/A

**Review Assessment: Checking Correctness Of Experiments:**

I assessed the sensibility of the experiments.

**Review Assessment: Thoroughness In Paper Reading:**

N/A

---

> ### Author Response · Authors · 2019-11-14
> **Thank you for your helpful comments!**
>
> >  * Implicit in this paper is that any information outside the mask is a distractor and any information inside is not a distractor....
>
> Yes, because many users of ML systems want the model to imitate human-level performance, i.e., look at the kinds of things that people would look at. So here we are adding prior knowledge about what a doctor would want to look at to make a decision, evaluate whether this is happening, and the impact on classification performance.
>
> >  The paper's novelty is somewhat limited. The idea of regularizing using saliency maps has been explored and even applied to medical data like the MSD used here in Gradmask ...
>
> We would like to point out that the original Gradmask reference is a workshop paper, and this is the first paper presenting that approach in detail. We used MSD for consistency: https://openreview.net/forum?id=Syx2z2aMqE
>
> >  Activation matching is also common (e.g. [1]), though it has not been combined with masking before.
>
> Thank you for this paper, we didn’t realize that our method had the same construction as $L_{content}$ in this paper, and we have now cited it (Page 3). Our method uses the same image twice (one masked), which means the model does not learn any representations from regions outside of the mask.
>
> >  While the weights applied to the various losses are provided, it's not clear how they were tuned...
>
> Extensive hyperparameter searches were run for all baselines (e.g., Gaussian blur, learning rate) as stated in the text, and the same hyperparameters were used for all new methods, with no tuning done on their specific lambdas (i.e., the gradmask lambda, the Actdiff lambda). We agree that our new models would have achieved higher test scores if we ran full hyperparameter searches for each new proposed model, but we cannot run these experiments during the rebuttal period (due to the number of configurations presented and the number of seeds averaged over).
>
> >  The results (and the conclusions) suggest Actdiff is not very effective at increasing generalization...
>
> Yes, while Actdiff clearly changes the saliency map dramatically (Figure 4, now updated to be much more comprehensive), it does not affect generalization performance, negatively or positively, in a consistent way. Given how interested people are in using saliency maps for model interpretability and for detecting overfitting, we believe this is an interesting empirical result.
>
> >  Missing experiments: In the Multi-Site experiment, compare to what happens when the circular mask is applied to all images and not just those from one site...
>
> This was the experiment that we performed. The mask was applied to all images in the training set, which drew examples from both sites. We have made this clearer in the paper (Page 8).
>
> > Presentation weaknesses: In the synthetic dataset the model cannot tell the difference between correct and incorrect signals at train time...
>
> The model always has access to the correct predictive signals (the crosses), so all models could learn the task if it also built features using the crosses. However, due to the dynamics of gradient descent, this never happens in practice without the actdiff regularizer.
>
> >  What is the Conv AE? I assume it is a CNN based autoencoder of some sort...
>
> Yes, this is specified at the bottom of page 4, which is now more explicit. The full model is detailed in section B of the appendix (page 11).
>
> >  What are the lambda hyperparameters?..
>
> Thank you for pointing out this oversight. They are indeed the weights of the loss terms, which we now make explicit.
>
> >  How does f(.) relate to the function o_l(.)?..
>
> f(.) the encoder of all models. o_l(.) is the preactivation outputs of layer l in f(.). This is general since an encoder is used in all experiments. This has been made clearer in the text (Page 3).
>
> >  * The MSD dataset is not clearly described...
>
> Thank you, please see revised text (Page 5).
>
> >  I think only Conv AE and UNet contain reconstruction losses...
>
> Yes, this is right. We now explicitly state "All auto-encoder models presented in this paper employed [$L_{recon}$].” (Page 3).
>
> > For me this paper requires special scruitiny because it presents a negative result.  ... Overall I think this paper is close but fails to meet the bar because it does a bit worse than expected on most criteria above.
>
> We believe you like this paper, but are _underwhelmed_ with our execution. We believe we addressed all the concerns you raised except the issue of not extensively tuning the hyperparameters on the new methods we proposed. We share your disappointment that we did not have the resources to run these experiments, but are currently running them and would be happy to include these results in the camera-ready version. We stand by our finding that the regularizers (particularly actdiff) control the saliency maps, so we feel our conclusion is nevertheless justified ("controlling saliency is a bad target for improving generalization performance").

---

### Decision · Program_Chairs · 2019-12-19

**Decision:**

Reject

**Comment:**

This paper studies the effect of training image classifier with masked images to exclude distraction regions in the image and avoid formation of spurious correlation between them and predicted labels. The paper proposes actdiff regularizer and demonstrates that it prevents such overfitting phenomenon on synthetic data.  However, there was no success on real data. This is important as it shows that the improvement reported in some saliency-map based approaches in the literature may be due to other regularization effects such as cutout.

This was a unique submission in my batch, as it embraces its negative results. Among our internal discussions, all reviewers that and we all believe that negative results are important and should be encouraged. However, in order for the negative results to be sufficiently insightful for the entire community, they need to be examined under well-organized experiments. This is the aspect that the reviewers think the paper needs to improve on.  In particular, R2 believes the paper could consider a larger set of possible regularizations as well as a broader range of  applications. The insights in such setting may then lead to solid insights on why the current approaches are not very helpful, and in which direction the follow-up researches should focus on.